# Photoperiod controls vegetation phenology across Africa

Tracy Adole [1*], Jadunandan Dash[1], Victor Rodriguez-Galiano[2] & Peter M. Atkinson[1,3,4]

Vegetation phenology is driven by environmental factors such as photoperiod, precipitation, temperature, insolation, and nutrient availability. However, across Africa, there's ambiguity about these drivers, which can lead to uncertainty in the predictions of global warming impacts on terrestrial ecosystems and their representation in dynamic vegetation models. Using satellite data, we undertook a systematic analysis of the relationship between phenological parameters and these drivers. The analysis across different regions consistently revealed photoperiod as the dominant factor controlling the onset and end of vegetation growing season. Moreover, the results suggest that not one, but a combination of drivers control phenological events. Consequently, to enhance our predictions of climate change impacts, the role of photoperiod should be incorporated into vegetation-climate and ecosystem modelling. Furthermore, it is necessary to define clearly the responses of vegetation to interactions between a consistent photoperiod cue and inter-annual variation in other drivers, especially under a changing climate.

[1] School of Geography and Environmental Science, University of Southampton, Southampton SO17 1BJ, UK. [2] Physical Geography and Regional Geographic Analysis, University of Seville, Seville 41004, Spain. [3] Faculty of Science and Technology, Lancaster University, Lancaster LA1 4YR, UK. [4] School of Geography, Archaeology and Palaeoecology, Queen's University Belfast, Belfast, BT7 1NN Northern Ireland, UK. *email: tracyadole@gmail.com

Vegetation phenology, the seasonal growing cycle of plants, is known to be sensitive to seasonal variation in environmental drivers such as precipitation, temperature, solar insolation, photoperiodic signals, and nutrient availability[1–3]. At the same time, changes in vegetation phenology can themselves influence some of these variables, a phenomenon known as vegetation phenological feedback[4]. Thus, long-term observation of vegetation phenology can serve as a suitable means of monitoring ecosystem responses to climate variability and change[5].

Globally, numerous studies, both using ground-based observation and satellite remote sensing-based data have investigated the key drivers of vegetation phenological responses[4,5]. These studies have reported that in mid and high latitudes, vegetation phenology is controlled mostly by temperature and photoperiod[1,6], while it is widely accepted that precipitation (water availability) is the dominant controlling effect on vegetation phenology in the tropics[7,8]. These factors can affect vegetation phenology together at the same time of year or separately at different times of year[1,2], and in a changing climate they can alter the typical timing of phenological events. This has been observed across the northern hemisphere where warmer spring temperatures resulted in an earlier start of season and delayed end of season, leading to longer growing seasons[9].

While there have been many studies on the environmental drivers on vegetation phenology in the northern hemisphere, few detailed studies quantifying vegetation phenological responses to these environmental drivers have been undertaken in Africa[10,11]. Indeed, the triggers of onset of vegetation growth and the beginning of dormancy in Africa are poorly understood. There exists much ambiguity about the environmental drivers of vegetation seasonal growth and pattern, for example: the large pre-rain green up of plants in Africa[12,13], irregularities in the relationship between onset of growing season and the beginning of rainy season in the Sahara desert[14], and uncertainty in the drivers of changing vegetation greenness in the Sahel[15]. These uncertainties may have contributed to the observed large biases in atmospheric components (mean temperature/precipitation/solar radiation) of ecosystem models which consequently lead to imperfect predictions of global warming impacts on terrestrial ecosystems[16]. This is because as much as 38% of the global climate-carbon cycle feedback comes from Africa[17]. Therefore, it is important to properly characterise Africa's vegetation phenological responses to environmental cues for accurate parameterisation of global land–atmosphere models.

Apart from Guan et al.[18], who found that insolation controls tropical evergreen vegetation development, studies investigating the relationship of Africa's vegetation phenology with environmental factors other than precipitation are scarce. Moreover, most of these studies were undertaken at the regional or country levels. For example, Van Rooyen et al.[19], discovered that rainfall is effective in the growth pattern of some woody species in southern Africa only when certain thresholds of temperature and photoperiod have been exceeded. Njoku[20,21] also found that vegetation seasonality in some rainforest species in Nigeria is not marked by seasonality in rainfall, but by seasonal changes in day length.

This multifaceted association of vegetation phenology with environmental drivers has not been investigated for the entire African continent. As a result, the sensitivity of the precipitation-controlled African phenology to other environmental drivers is not well understood. This is particularly important in the context of climate change; for example, the effect of projected changes in precipitation on the pre-rain green-up of plants[12], and the recently observed lengthening of the growing season in western Africa[22].

Also, large biases and failures of ecosystem models have highlighted the need to re-evaluate the drivers of vegetation growth and senescence[16]. Studies have also shown that more accurate representation of vegetation phenological processes in these models have improved the models' climate change predictions and sensitivities[23]. As a result this research aims to provide greater understanding of the climate-driven vegetation phenology of Africa by undertaking a systematic analysis of the relationship between remotely sensed vegetation phenological parameters (also known as land surface phenology, LSP) and a range of climatic drivers. Since several studies have shown that certain thresholds of temperature, insolation and water availability[6,24] are needed to be attained before vegetation growth or dormancy is initiated, "preseason" climatic drivers were used for this research. The preseasons are well-defined periods before phenological dates. We, therefore, aim to quantify the LSP responses to these preseason climatic drivers spatially and, more specifically, determine the *dominant* climatic driver of the LSP of the major land cover types in Africa.

Here, we demonstrate that photoperiod is a more important environmental cue than previously thought, while also acknowledging the significance of other factors. The results support the proposition that vegetation phenology is influenced by the combination of two or more factors rather than a single factor. It also supports previous ideas that LSP response is biome-dependent. These findings enhance our understanding of how LSP will respond to variation in environmental drivers under a changing climate. They also highlight the importance of incorporating photoperiod and temperature settings when developing phenological models for tropical regions such as Africa.

## Results

**Correlation between Start of Season (SOS) and climatic factors.** Differences were observed in the associations between the start of season (SOS, in Day of Year units) and climatic drivers which may be dependent on the vegetation type or the geographical region. The correlation analysis showed that photoperiod had the largest significant correlation with SOS in all vegetation types and geographical regions (Figs. 1–3, and Supplementary Tables 1–3), implying that SOS is mainly controlled by photoperiod in Africa. This large correlation was not as a result of latitudinal influence as might be hypothesised (since latitude is a major factor in computing daylength; see Supplementary Figs. 1 and 2). Partial correlation results (Supplementary Tables 1–3) revealed that a combination of other climatic factors also exert substantial effects on SOS date.

The SOS of all studied vegetation types in the northern region of Africa were significantly and positively correlated with photoperiod in both statistical models (Fig. 1, Supplementary Table 1). These results suggest that the longer the hours of daylight the later the SOS date. The partial correlation analysis revealed that a combination of these factors tends to increase the correlation for all climatic drivers, especially in the 30 and 40 days preseason periods (Supplementary Table 1). The results also showed that the amount of preseason rainfall was significantly and positively correlated with the SOS of croplands, grasslands and woodlands except for the first two preseason periods of shrublands, which had insignificant correlations. Correspondingly, preseason temperature was significantly and negatively correlated with SOS of croplands, grasslands and shrublands with the exception of woodlands which produced small positive correlations. For preseason solar radiation averages, all preseason periods for shrublands, and the first three periods for croplands were generally not significantly correlated with SOS. In contrast, preseason solar sums and SOS were negatively correlated in grasslands, and positively correlated in woodlands.

Results in the southern hemisphere of Africa are similar to those in the northern hemisphere except for major differences in croplands and shrublands (Fig. 2, Supplementary Table 2). In

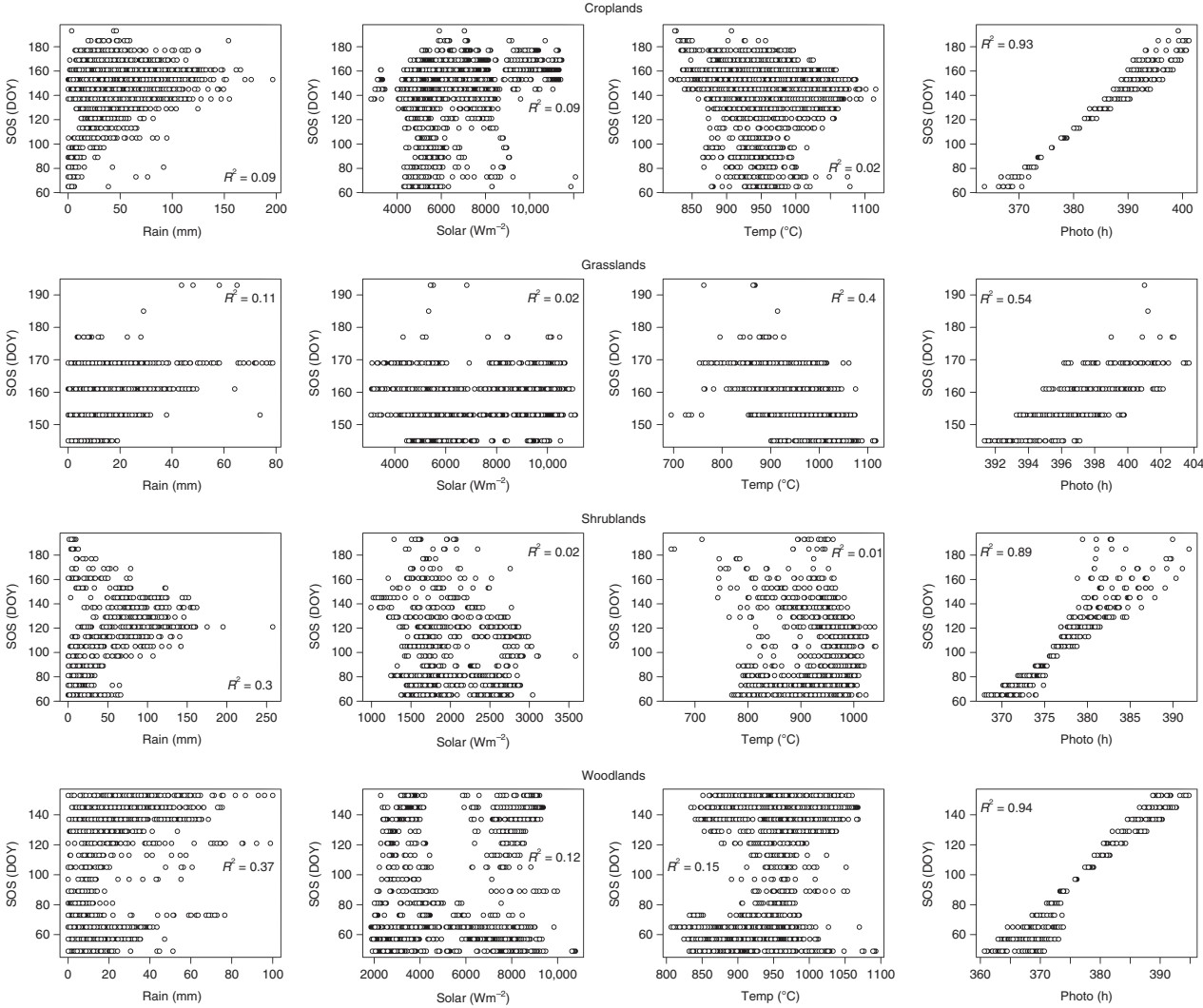

**Fig. 1** Scatterplots between SOS and climatic drivers cumulated over a 30 day preseason period across different vegetation types in the Northern hemisphere of Africa (All at $P < 0.05$). Plots for other preseason periods are not shown (see Supplementary Table 1). Shrublands were located in the horn of Africa

both models, SOS was positively correlated with photoperiod, implying that later vegetation onset dates in the southern hemisphere relate strongly with longer hours of daylight. However, exceptions were SOS dates for croplands in the South-western region, which had much earlier SOS dates, similar to those in western Africa but negatively correlated with photoperiod. In the same way, SOS dates in the extreme north of Africa for croplands and shrublands were significantly and negatively correlated with photoperiod. These outcomes suggest that for preseason periods with longer hours of daylight, decreases in daylength are associated with earlier SOS dates, while for preseason periods with shorter hours of daylight, increases in daylength are associated with earlier SOS dates (Fig. 4). For preseason temperature averages, significant negative correlation values were observed in preseason periods of 10–30 days for croplands and grasslands. For shrublands and woodlands, significant negative values were observed in preseason periods of 60–80 days. These significant negative correlations suggest that earlier SOS dates were associated with higher preseason temperatures. For the significant correlations between preseason solar averages and SOS dates, correlation values were small, suggesting that solar radiation may not have a substantial effect on onset of vegetation growth in these regions.

**Correlations between EOS and climatic factors**. Contrary to the results for SOS, the statistical analysis for EOS revealed the influence of factors which, in addition to photoperiod, are major factors controlling EOS dates in Africa (Figs. 5–7, Supplementary Tables 4–6). For example, for shrublands, SOS had larger correlation values than any of the preseason climatic factors, suggesting that EOS dates are largely controlled by the date of onset of vegetation growing season. In the northern region, while preseason photoperiod was most important for croplands, grasslands and shrublands, in woodlands preseason rainfall had the largest correlation values (Supplementary Table 4). The preseason rainfall was significantly negatively correlated with EOS, suggesting that the greater the amount of preseason rainfall the earlier the EOS dates. Although significantly negative, the preseason photoperiod correlation values were small, suggesting that the duration of sunlight may not have a great effect on senescence of woodlands in northern Africa. However, for croplands and grasslands, the results showed preseason photoperiod as the most important controlling climatic factor. For croplands EOS was significantly and positively correlated with photoperiod. In contrast, for grasslands and shrublands preseason photoperiod was significantly and negatively correlated with EOS. These differences may be due to the length of growing season of croplands,

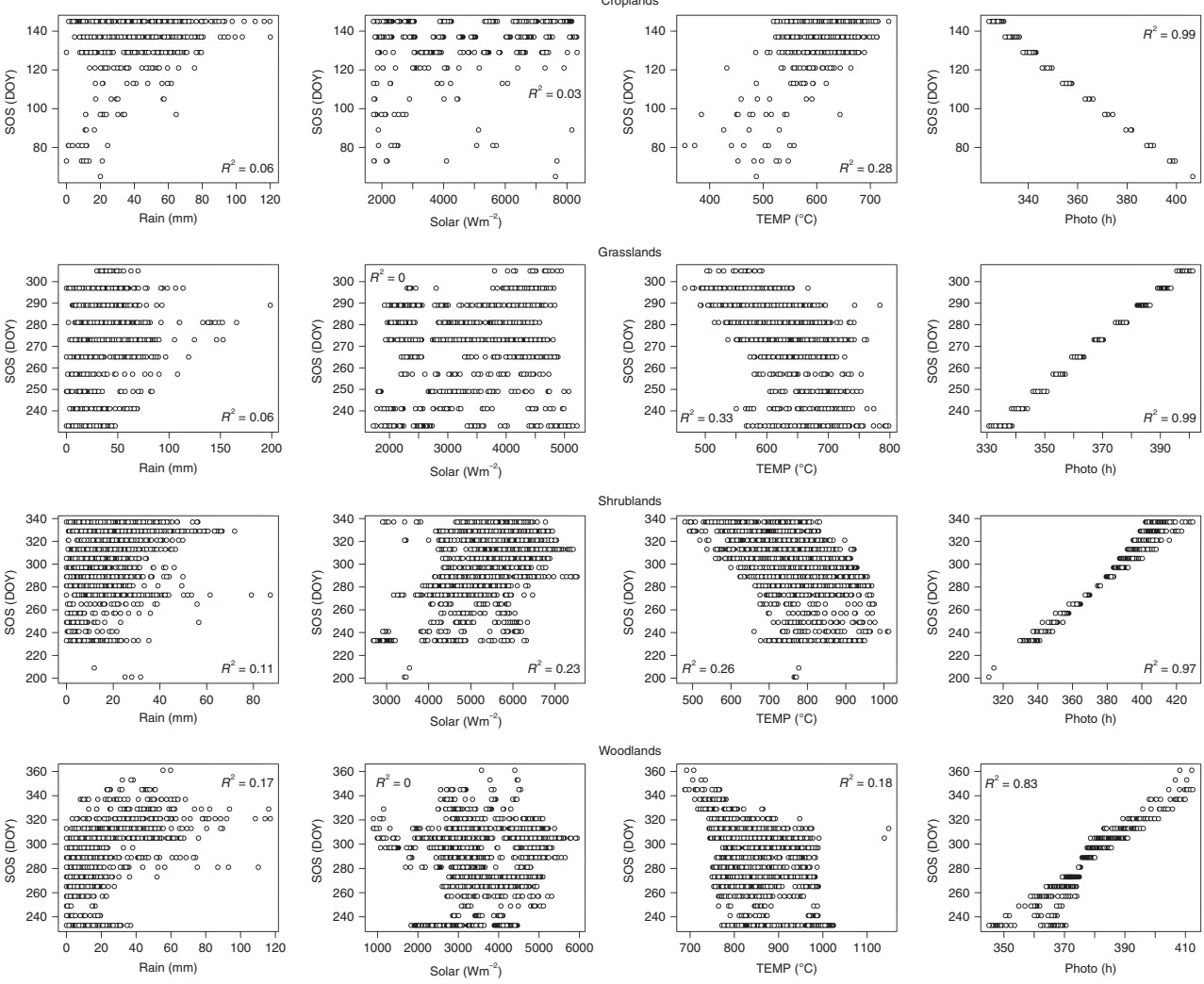

**Fig. 2** Scatterplots between SOS and climatic drivers cumulated over a 30 day preseason period across different vegetation types in the Southern hemisphere of Africa (All at $P < 0.05$). Plots for other preseason periods are not shown (see Supplementary Table 2). Croplands were located in the south-western region of Africa with a similar climate to the Sudano-Sahel region of western Africa

and the photoperiod seasonality in the Sudano–Sahel region, which increases at the start of the year, peaks in the middle and declines towards the end of the year.

As previously mentioned, preseason temperature, like photoperiod, plays an equally important role in EOS onset in the southern hemisphere (Supplementary Table 5), and is the primary factor in the extreme north of Africa (Supplementary Table 6). The exception to this is preseason temperature for croplands in south-western Africa. Partial correlation analysis revealed that although preseason temperature was significantly and negatively correlated with EOS, correlation values in all preseason periods were less than that of photoperiod. For grasslands, shrublands and woodlands in southern Africa, preseason temperature was significant and positively correlated with EOS, suggesting that increases in preseason temperature may delay EOS dates. On the other hand, preseason temperature for croplands and shrublands in the extreme north of Africa was significantly and negatively correlated with EOS, suggesting that increases in preseason temperature may result in earlier EOS dates (Supplementary Table 6). These negative correlations at very high latitudes and positive correlations at lower latitudes suggest that preseason temperature increases at higher temperatures contribute to earlier EOS while temperature increases at lower temperatures contribute to later EOS.

The results for preseason photoperiod were somewhat different from those of temperature. In the southern hemisphere, for grasslands, shrublands and woodlands, increasing preseason photoperiod was associated with earlier EOS dates, while in the extreme north, increasing preseason photoperiod was associated with later EOS dates.

## Changes in associated climatic factors and inter-annual variation in LSP

As expected, there was no change in inter-annual trends of the preseason photoperiod estimated from median SOS and EOS values over the time period, as photoperiod is consistent from year-to-year. Also, significant inter-annual trends in preseason precipitation and solar radiation were found to occur with no particular spatial pattern, and no association was found with significant changes in LSP dates. On the other hand, preseason temperature for SOS and EOS dates revealed significant increasing inter-annual trends with an average slope of 0.33 °C/year, and mostly in the Sahel region overlapping with areas showing significant trends in LSP dates over the time period (Fig. 8). This suggests that changes in preseason temperatures may have influenced observed inter-annual variation in LSP dates.

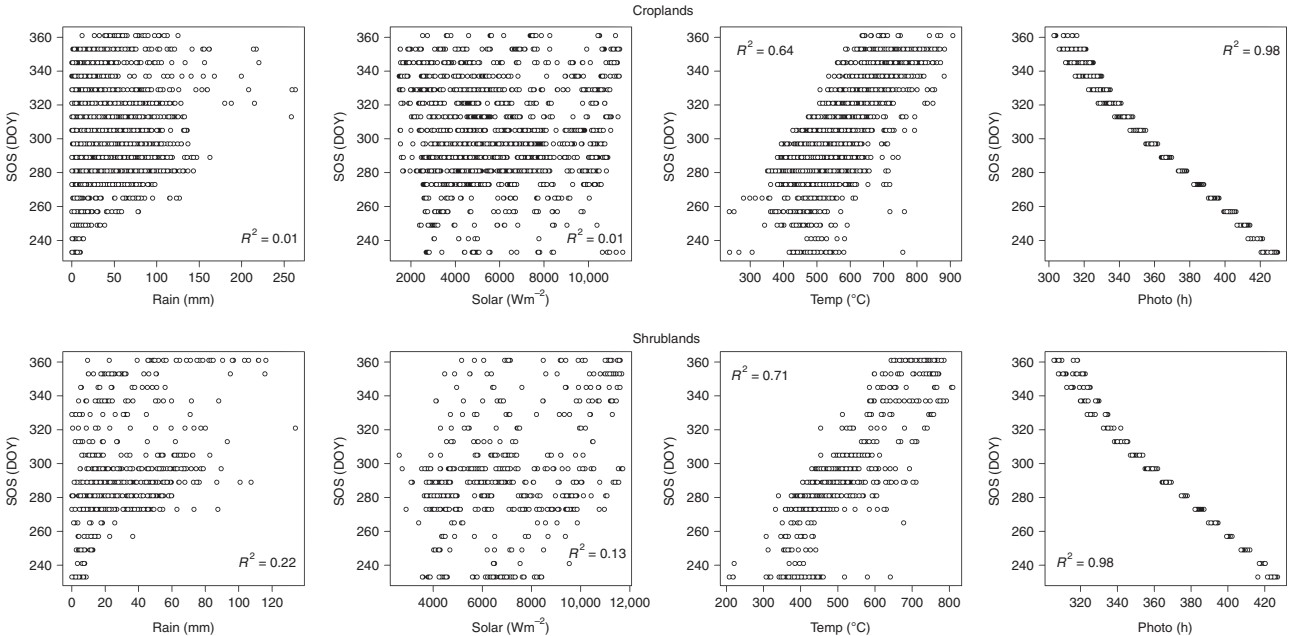

**Fig. 3** Scatterplots between SOS and climatic drivers cumulated over a 30 day preseason period across different vegetation types in the extreme northern part of Africa (All at $P < 0.05$). Plots for other preseason periods are not shown (see Supplementary Table 3)

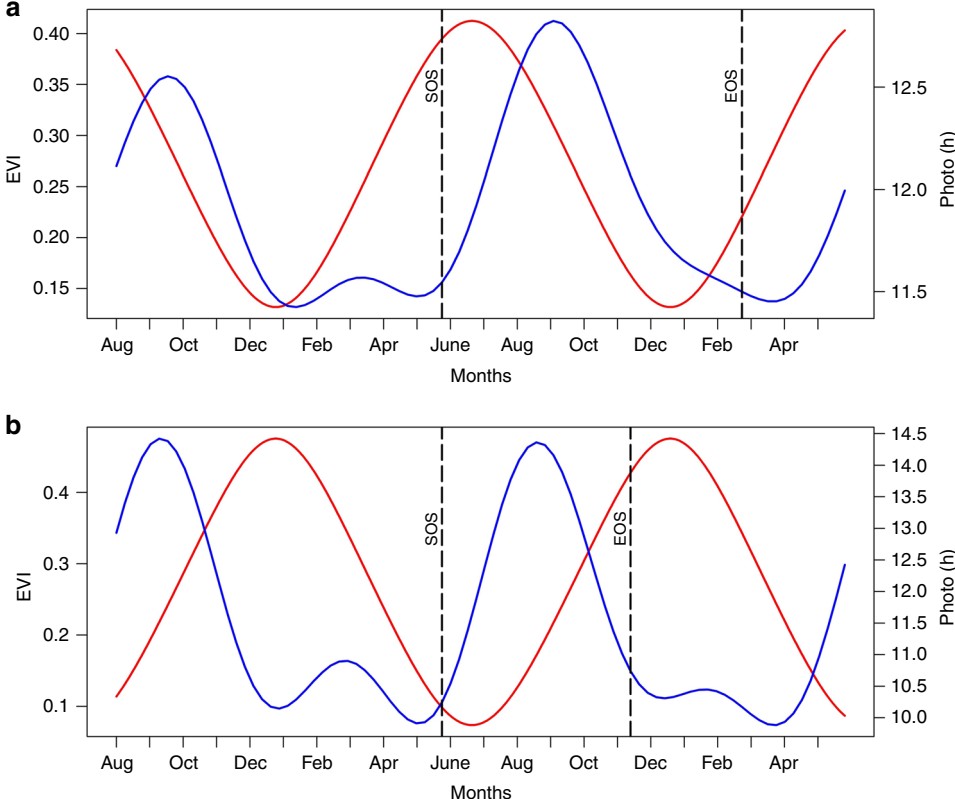

**Fig. 4** Example of pixel profile for a complete EVI and Photoperiod daily time-series: **a** Croplands in the Sudano-Sahel region showing increase in (longer) growing season with increasing preseason photoperiod at the start of vegetation growing season (i.e., synchrony), **b** Croplands in the south western region showing increase in (shorter) growing season with decreasing preseason photoperiod at the start of vegetation growing season (i.e., asynchrony). EVI time-series is represented by blue curves while photoperiod is represented by red curves. Vertical dashed black lines show LSP parameters (SOS and End of Season; EOS)

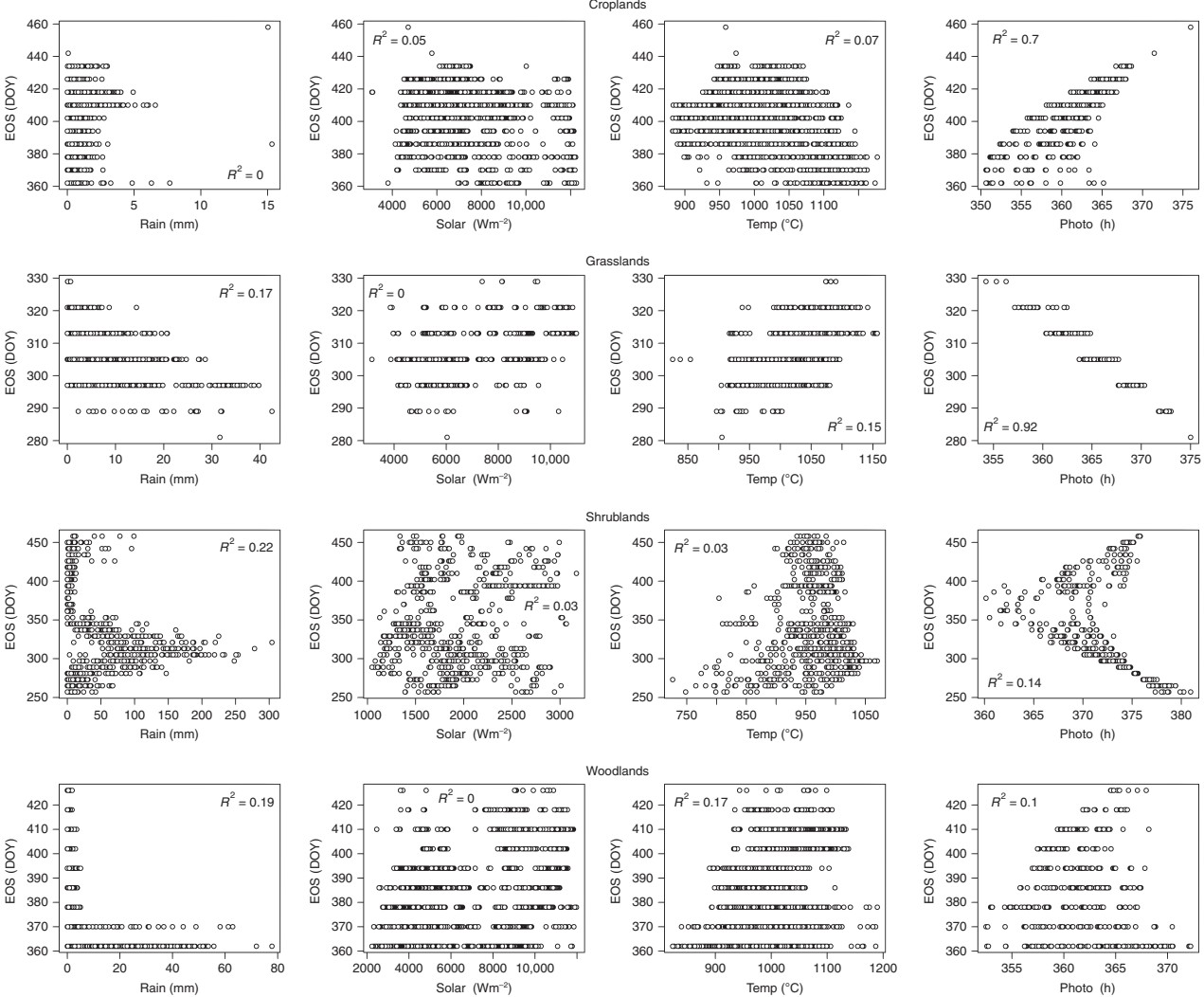

**Fig. 5** Scatterplots between EOS and climatic drivers cumulated over a 30 day preseason period across different vegetation types in the Northern hemisphere of Africa (All at $P < 0.05$). Plots for other preseason periods are not shown (see Supplementary Table 4). Shrublands were located in the horn of Africa

## Discussion

Across Africa, photoperiod was found to be the dominant factor controlling the onset and end of vegetation growing season. This highlights the high sensitivity of plants to photoperiod, a phenomenon that has been documented as early as 1950s[20,21,25,26]. This dominant control by photoperiod tends to corroborate earlier work which attributes photoperiod relative to some threshold to be the major determinant that allows other climatic-driven development to occur[27]. It also provides more evidence supporting the pre-rain green-up phenomenon observed in Africa[12], further challenging the widely held belief that onset of vegetation growing season in Africa is water limited. The results are also in agreement with earlier studies indicating that a combination of climatic factors, either occurring simultaneously or preceding one another, controls LSP patterns, with their effects sometimes biome-dependent[1,2,28]. It further supports the idea of incorporating photoperiod into terrestrial biosphere models for increased accuracy of prediction[29,30].

In the northern latitudes of Africa, photoperiod and temperature were found to be the major climatic factors controlling the onset and end of vegetation growing season. This result for Africa is nevertheless consistent with other research which concluded that photoperiod is the major factor controlling phenological events in tropical ecosystems[20,31].

In the extreme north of Africa, the wet season is usually accompanied by a declining daylength duration and an increasing temperature. With this research revealing negative correlations between SOS and preseason photoperiod, and positive correlations between SOS and preseason temperature, it can be inferred that a combination of lower temperature limits and higher photoperiod limits are the cues required for the initiation of vegetation growth in the extreme north of Africa. These findings are similar to typical vegetation phenology drivers for northern hemisphere[1,6]. For onset of dormancy, the result suggests that the reverse (higher temperatures and lower photoperiod) may be the environmental cue, with temperature playing a more dominant role.

In the Sahel region, onset of growing season for all studied vegetation types is predominantly controlled by photoperiod, suggesting its strong sensitivity to sunshine duration. This sensitivity has been reported for a wide range of vegetation types in the Sudanian region[32] and is also known to be genetically based in some cereal crops, including major varieties grown in West Africa[33,34]. Nonetheless, in this region, the role of other factors and their combinations are still important as shown in the results. For example, the onset of vegetation growing season is characterised by increasing daylength duration and increasing temperatures. The photoperiod seasonality from the beginning of the

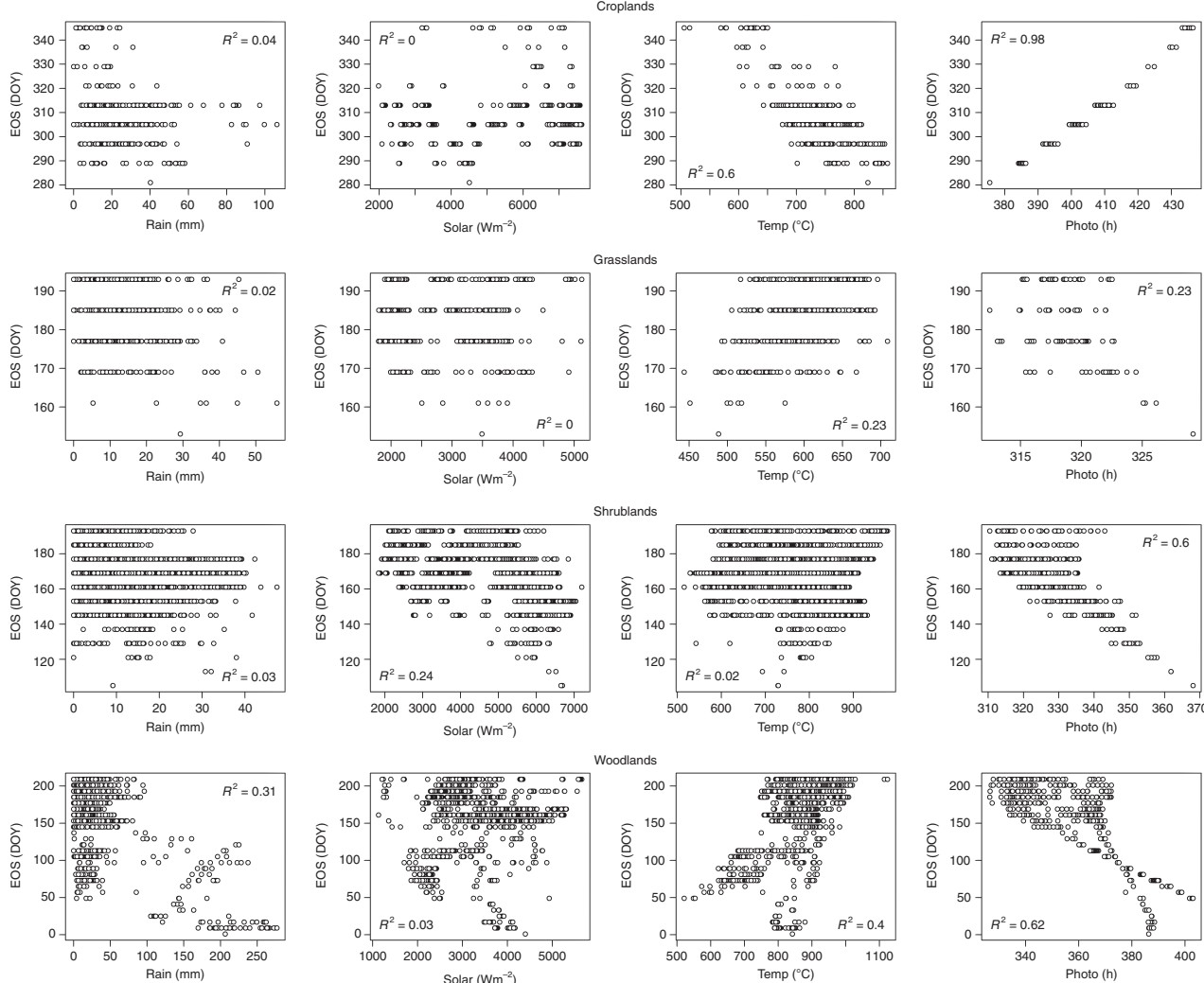

**Fig. 6** Scatterplots between EOS and climatic drivers cumulated over a 30 day preseason period across different vegetation types in the Southern hemisphere of Africa (All at $P < 0.05$). Plots for other preseason periods are not shown (see Supplementary Table 5). Croplands were located in the south-western region of Africa with a similar climate to the Sudano-Sahel region of western Africa

year usually begins with longer day length of over 11 h, and rising temperatures of over 20 °C. With both factors having significant correlations (photoperiod with the largest), these observations suggest that lower photoperiod limits and warmer temperatures (negative correlation of preseason temperatures), coupled with the timing of the onset of rainy season could be responsible for the initiation of vegetation green-up in this region. These conditions are mostly favourable to tropical plants, known to grow well in warmer temperatures and with shorter photoperiods[28], particularly millet and sorghum, two of the most important cereal crops grown in Sudano–Sahelian region[35]. Surprisingly, preseason rainfall had little or no significant effect on onsets dates, suggesting that the amount of precipitation plays a secondary or no role when compared to photoperiod and temperature in regulating the start of vegetation growing season in the Sahel. This is supported by earlier studies which reported a large percentage of pre-rain green-up has been observed in this region[12]. In addition, contrary to expectations, significant positive correlations were observed between preseason rainfall and SOS dates. A possible explanation for this might be that greater amounts of rainfall are usually accompanied by clouds, thus, reducing temperatures and sunshine intensity below growth initiation thresholds, hence resulting in a later onset dates[36].

For the onset date for grasslands, smaller positive correlations with preseason photoperiod and a slightly greater negative correlation of solar radiation and temperatures were observed in this region, compared to other land cover types. Studies have shown that for grasses (mainly of C4 type) found in many African ecosystems phenology is driven by high solar radiation and temperatures[37]. It has been established that C4 plant types are better adapted to warm climates because of their enzyme sensitivity to chilling temperatures[38]. They are also known to have greater photosynthetic capacity at higher sunlight and temperature levels[39]. These factors may explain the relatively significant negative correlation of solar radiation and temperature with the onset dates of grasslands. This further supports the recommendation that thermal scenarios should be considered when investigating grassland phenology[40], since increased amount and duration of rainfall had no effect on its phenology events. These findings in general, raise the likelihood of a vegetation type dependency of LSP responses to climatic factors. Additionally, it also highlights the much reduced role of rainfall seasonality in the vegetation growth cycle. However, it is important to note that although most studies have shown a large correlation between rainfall and vegetation seasonality, this association is more related to timing (onset of raining reason and onset of vegetation

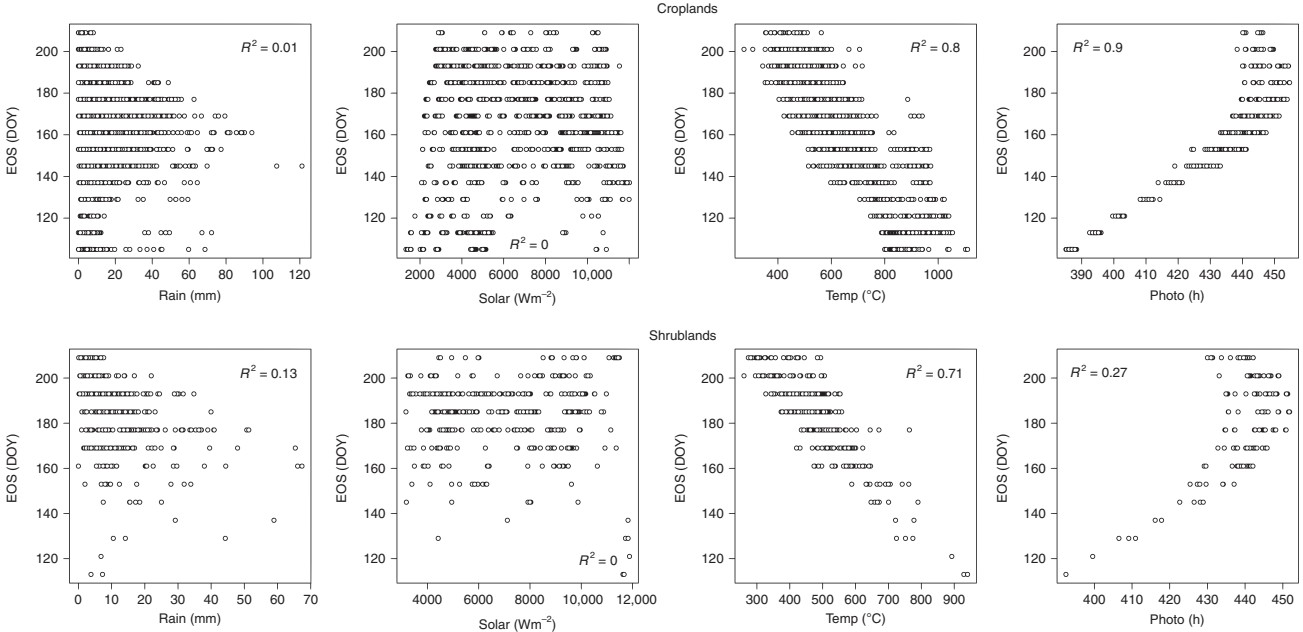

**Fig. 7** Scatterplots between EOS and climatic drivers cumulated over a 30 day preseason period across different vegetation types in the extreme northern part of Africa (All at $P < 0.05$). Plots for other preseason periods are not shown (see Supplementary Table 6)

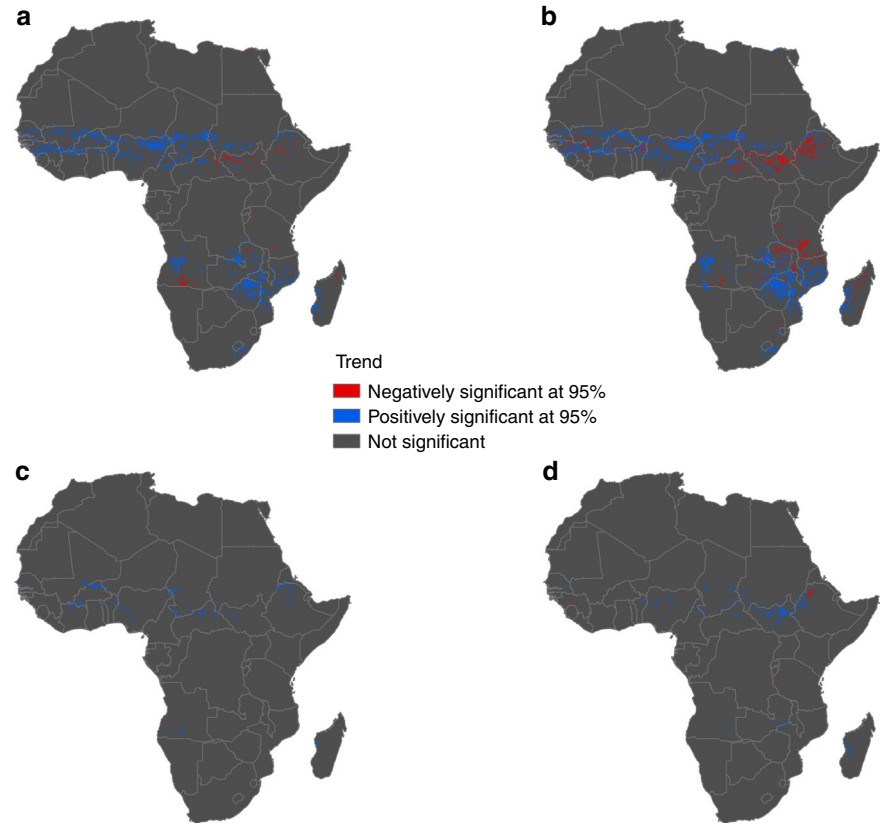

**Fig. 8** Spatial distribution of significant inter-annual trends of LSP parameters and the associated 30 days preseason cumulative temperature from 2001 to 2015. **a** Significant SOS trends, **b** significant EOS trends. **c** Pixels of significant trends in preseason temperature of SOS which overlap with significant trends in SOS dates. **d** Pixels of significant trends in preseason temperature of EOS which overlap with significant trends in EOS dates

growing season) than to pre-seasonal rainfall amounts. As shown in this research, the amount of rainfall has little or no significant influence on onset dates. Rather, the association is largely a time-based relationship as shown in previous studies[12,41].

Unlike the onset of vegetation growing season, for dormancy onsets, preseason photoperiod was not the only major determining factor. While preseason photoperiod was the predominant factor controlling dormancy onsets in croplands and

grasslands, other factors were shown to be more significantly associated with EOS dates. In shrublands, SOS and preseason rain were more dominant, and in woodlands, preseason rain and solar radiation showed more dominance, although their effect is dependent on the preseason period. This negative correlation of preseason rainfall could be caused by the accompanying reduced temperatures not favourable to vegetation growth as explained above.

Apart from the human factor in agricultural lands (irrigation/farmers' decision of sowing dates)[2,33], the length of growing season as a function of the vegetation type could also be a factor that can contribute to the effect of preseason climatic factors on EOS dates. For example, in croplands preseason photoperiod was significantly positive, whereas that of grasslands was significantly negative. In croplands the length of growing season extends to periods at the beginning of the year where there is a small but increasing photoperiod. While, in grasslands with a short growing season of approximately six months, the photoperiod towards the end of the season is large but declining. Also, in shrublands, photoperiod had larger significantly negative correlation values in the preseason periods of 2–3 months before the onset of dormancy dates, suggesting that photoperiod 2–3 months before onset of dormancy plays a major role in regulating EOS in shrublands. Likewise important in determining EOS is the timing of the onset of growing season in shrublands (significant SOS values in shrublands).

Similarly, in the south photoperiod was the major climatic factor controlling onset of vegetation growing season while other factors showed significant control of vegetation dormancy onset. These findings are consistent with Garonna et al.[42] and corroborate the idea that photoperiod is the most reliable predictor of onset dates for southern African savanna trees[43,44]. Equally, the apparent positive effect of preseason photoperiod was as a result of higher and increasing preseason photoperiod. However, the observed negative correlation of preseason photoperiod on croplands in south-western Africa can be attributed to the declining duration of day length which is similar to the photoperiodicity of the extreme north of Africa. Also significant were preseason temperatures for grasslands and croplands which had a negative correlation, with warmer temperatures favouring earlier vegetation green up. These results confirm previous suggestions that a combination of photoperiod and temperature thresholds are environmental cues for vegetation growth in southern Africa[19].

Preseason rainfall amount had no effect on SOS, except for preseason periods in grasslands. This was expected as pre-rain green-up has been reported to be ubiquitous in southern African savanna, with as early as 60 days before the first rains[12,13]. In addition, these results seem to be consistent with other research which found that rainfall clearly had no effect on the development of leaves in some southern Africa savanna trees[43]. This further confirms that most of the associations between rainfall and vegetation seasonality are related mainly to time and productivity[45]. For example, the memory mechanism of miombo woodlands: greening-up in anticipation of onset of rains[46] (time-based), and the intra-seasonal rainfall variability effect on sorghum yield[47] (productivity-based).

The onset of vegetation dormancy was influenced not only by photoperiod but also by other climatic factors. In croplands, a positive correlation of preseason photoperiod was dominant, while in other studied vegetation types, preseason photoperiod had a negative influence. However, this negative influence of photoperiod is secondary to the positive influence exerted by preseason temperature in grasslands. Also in shrublands and woodlands, the influence of preseason temperature was significantly high, suggesting that temperature increases postpone

the onset date of vegetation dormancy. This observation is consistent with earlier studies which showed that increases in temperature may have extended the vegetation growing season in the Namaqualand, southern Africa[45]. In contrast, the effect of preseason rainfall suggests that increasing rainfall led to earlier onset date of vegetation dormancy. Again, the accompanied reduced temperature during rainfall could be responsible for the negative correlation[48].

In general, we observed an overall synchrony between photoperiod and LSP parameters across all of Africa, an observation supported by several studies highlighting photoperiod control of leaf flushing rather than rainfall[31,49]. A possible explanation for this may be due to the fact that photoperiod is the most consistent environmental signal from year-to-year[1,49]. As result of this consistency, plants may tend to rely more on specific day length signals to regulate their growth[34]. This can be seen in the results showing that increasing preseason photoperiods of above 12 h duration tend to be associated with later SOS and earlier EOS, while increases of above 10 h were associated with earlier SOS and later EOS. Similarly, decreasing preseason photoperiods of below 12 h were associated with earlier SOS and later EOS. Hence, it is possible to hypothesise that longer day length duration of above 12 h tends to delay the onset of vegetation growing season and initiate dormancy, while a duration of <12 h but above 10 h may initiate SOS and delay EOS. This suggest that a certain threshold of day length must be exceeded to initiate the onset of vegetation growing season, and in the same way initiate the end of vegetation growing season. This distinct change in the response of plants to small changes of 2 h or less in photoperiods has been reported previously[31]. Likewise, it has also been suggested that plants respond to specific critical daylength (varies from plant to plant) during which hormonal regulation of growth initiation or cessation hormones occurs[50,51]. This ability of plants to detect light and measure time very accurately has been attributed to an "endogenous time-keeping mechanism called the circadian clock"[50], and the perception of light signals by photoreceptors[3], and these clock and photoreceptors genes can be found in all living plant cells[51]. These responses to photoperiod have been shown to influence the population structure of major crops like millet and sorghum in Western and Central Africa than any other environmental factor, a sole factor for adaptation to environmental constraints[35,52]. However, there are still many unanswered questions about how this mechanism works with phenological parameters especially in different plant types, and more investigations are required as recommended by other studies[4,53].

A particular interesting observation which further supports the sensitivity of plants to small changes in photoperiods is the distinct response of croplands in the Sudano-Sahel region of western Africa and the croplands in south-western Africa. Although, estimated SOS beginning around May/June were similar for both croplands, however their responses to photoperiodic signals were very distinct. This can be attributed to the increasing preseason photoperiod at the start of the year observed in Sudano-Sahel region, and the decreasing preseason photoperiod observed at the same period in south-western Africa. This distinct response of crops to the direction of photoperiod also reflects the results from Nori et al.[54], who found out that leafing rate was determined by the duration and direction of photoperiod at seed germination. Also, the length of growing season as a function of the crop type may also play a role in these responses. For example, maize crop mostly grown in the south-western region have shorter growing season and harvested much earlier than those (millet/sorghum/cassava) in the Sudano–Sahel region[55].

Irrespective of the observed dominance of photoperiod, the partial correlation results also showed that LSP is influenced by a

combination of factors, which is in agreement with previous studies showing that most phenological phases are controlled by both photoperiod and temperature[1,2]. Additionally, significant trends in LSP dates were observed in the study time period[22], and only trends in preseason temperature showed reasonable spatial overlaps suggesting that increasing temperature may have influenced observed inter-annual trends in LSP dates. Therefore, further investigations into LSP response to interactions between a consistent photoperiod and inter-annual variation in climatic drivers, especially under a changing climate, is paramount. The importance of such interactions has been brought to the fore recently by other researchers[30,53]. Understanding such interactions would help in identifying the confounding drivers of the reported inter-annual variation in vegetation phenology in Africa[22], especially knowing that photoperiod is consistent from year-to-year. Nevertheless, this research highlights the important role of photoperiod in vegetation phenology. Hence, we suggest that photoperiod is a key factor which should be incorporated into all vegetation phenological models. This importance can be corroborated by Liu et al.,[30] and Migliavacca et al.,[29] who reported a significant improvement in vegetation phenology model performance and uncertainty reduction resulting from the integration of photoperiod.

Nevertheless, it is important to note here that the findings of this study have to be seen in light of some limitations. The insufficient or the complete absence of field observation data for validation, and the spatial variation effect of the photoperiod curve. It is practically impossible to carry out proper validation of LSP estimates across the continent because of this scarce availability of ground data. Also, the photoperiod curve which changes smoothly over Africa may have exerted some spatial effect in the statistical modelling results. Therefore, we suggest that future analysis should investigate the best approach that can compute preseason period and same time control for spatial variation.

Another limitation of this study relates to uncertainties in the remotely sensed data. These range from sensor degradation issues to saturation effects due to clouds and aerosols[56] that can affect the quality of the data and in turn the phenological variable. However, steps were taken to mitigate the effect of such limitations, including computing the EVI to reduce the effect of saturation.

In conclusion, our study revealed a predominately photoperiodic control on vegetation growth (SOS and EOS) across all of Africa. This provides evidence that, contrary to widely held expectation, rainfall is not a direct driver of vegetation onset and end dates in Africa. It showed that vegetation phenology is sensitivity to photoperiod. The onset and end dates were either significantly positively or negatively correlated with preseason photoperiod which is largely dependent on the seasonality of photoperiod, and synonymous with the wet and dry seasons in Africa.

## Methods

**Study area LSP and land cover datasets**. Africa is the second largest continent in the world covering an area of over 30 million km², and its latitude ranges from 37° N to 35°S with the Equator in the middle[57]. With a diverse range of vegetation types, as shown in Fig. 9, Africa is home to the world largest area of savannah, the second-largest rainforest in the world, and the second and third largest wetlands in the world[58]. Africa has a distinctive pattern of climate which varies by geographical sub-region and which plays a significant role in the vegetation dynamics of Africa[57]. See Supplementary Fig. 3 for spatial distribution of yearly mean values of climatic factors across the continent. Medium spatial and temporal resolution satellite sensor data provided by the Moderate Resolution Imagining Spectrometer (MODIS) sensor were used for this analysis. The specific datsets are the MODIS/ Terra Surface Reflectance 8-Day L3 Global 500 m data (MOD09A1), and MODIS/ Terra Land Cover Type Yearly L3 Global 500 m data (MCD12Q1). 16 years (18 Feb 2000–29 Aug 2015) of MOD09A1 and 13 years (2001–2013) of MCD12Q1 with collection numbers ranging from h16v05 to h22v11 were downloaded from NASA's LP DAAC (https://lpdaac.usgs.gov/).

**LSP estimation**. The Enhanced Vegetation Index (EVI), which was developed to address some of the limitations of NDVI[56], was calculated from the MOD09A1

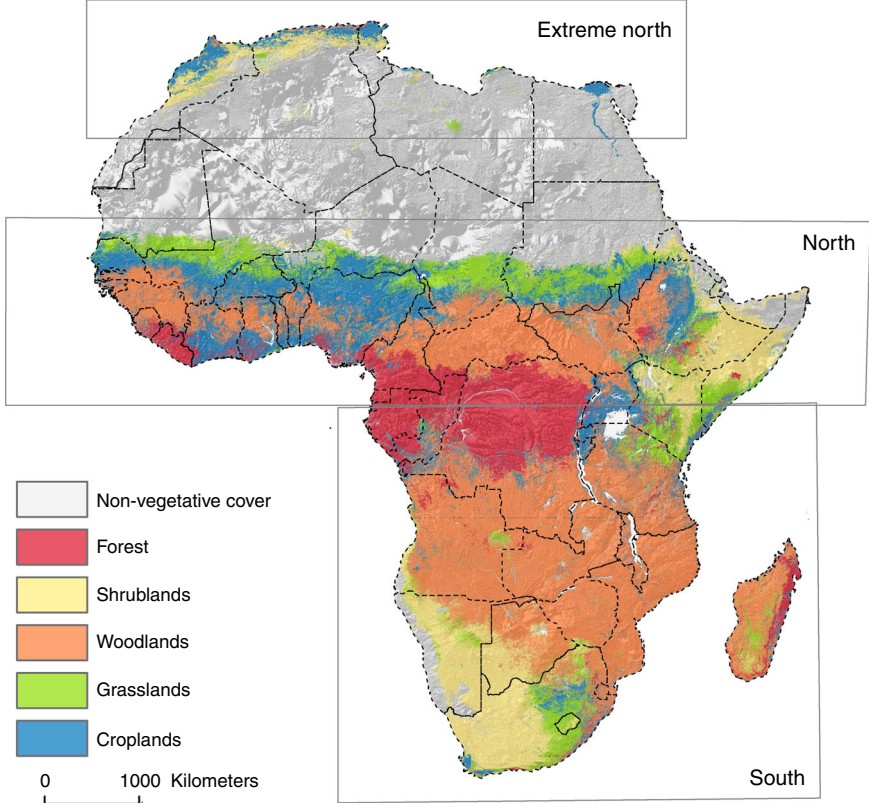

**Fig. 9** Reclassified land cover types and the different geographical study regions

**Table 1 LSP parameters and preseason climatic drivers used in this research showing their spatial resolutions and the data sources**

| Parameters | Symbol/Units | Datasets | Resolution | Source |
|---|---|---|---|---|
| Start of season | SOS/DOY | Land Surface Phenology (LSP) parameters for Africa | 500m | https://doi.org/10.5258/SOTON/D0407 |
| End of season | EOS/DOY | Land Surface Phenology (LSP) parameters for Africa | 500m | https://doi.org/10.5258/SOTON/D0407 |
| Preseason cumulative rainfall | Rain (mm) | CHIRPS | 0.05° | https://www.chg.geog.ucsb.edu/data/chirps/ |
| Preseason cumulative solar radiation | Solar (W m$^{-2}$) | ERA-Interim | 0.125° | https://www.apps.ecmwf.int/datasets/ |
| Preseason cumulative temperature | Temp (°C) | ERA-Interim | 0.125° | https://www.apps.ecmwf.int/datasets/ |
| Preseason cumulative photoperiod | Photo (h) | | 0.125° | Grid created from 'geosphere' package in R (Hijmans, 2017) |

data using equation 1 after residual atmospheric and sensor effects were filtered out in a Quality Assurance (QA) assessment procedure detailed in the MODIS Land Products Quality Assurance Tutorial, 2016[59].

$$EVI = G * \frac{(NIR - Red)}{(L + NIR + C1 * Red - C2 * Blue)} \quad (1)$$

A time-series cycle of two years (i.e., six months before and after the target year) comprising 86 layers of EVI stacked data was used to estimate the LSP parameters[60]. This two years of data was used to account for non-uniform growing seasons across Africa[61]. As explained in Adole et al., and Dash et al.,[61,62], the Discrete Fourier Transform (DFT) technique was used to estimate the Start of Season (SOS) and End of Season (EOS) parameters for each year, for each image pixel covering the period 2001–2015. This technique is particularly useful in detecting the bimodal seasonality and double cropping agricultural system found in some parts of Africa. However, only the first season was considered in this analysis due to the very limited occurrence of the second season and limited spatial coverage of the corresponding climatic data.

To stratify the estimated phenological parameters based on land cover, the MODIS land cover data, which have been shown to perform relatively well when compared to other global land cover data, were acquired for this study[63]. Homogeneous pixels which were stable through the entire 13 year time-series were extracted from the MODIS land cover data due to the highly dynamic changes in Africa's land cover[64]. These homogeneous areas of each land cover type were further reclassified following the method in[12] and then used to mask out the LSP estimates of four major land cover types: croplands, grasslands, shrublands and woodlands.

**Climatic datasets**. The 0.05° gridded rainfall Climate Hazards Group InfraRed Precipitation with Station (CHIRPS) dataset, which has been shown to out-perform most other rainfall products in data-sparse regions in Africa[65], was selected for this research. It was generated by combining satellite sensor and station data using smart interpolation techniques[65]. 16 years (2000–2016) of daily rainfall data were downloaded from CHIRPS (http://chg.geog.ucsb.edu/data/chirps/).

Owing to the sparse availability of air temperature data over Africa, daily skin temperature data from 2000 to 2016 with a spatial resolution of 0.125° were acquired from the ERA-Interim developed by the European Centre for Medium-Range Weather Forecasts (ECMWF) (http://www.apps.ecmwf.int/datasets/). These data, unlike surface temperature data, represent the interface between the soil and atmosphere, and were generated using a series of improved data assimilation techniques (4D-Var analysis) and simple empirical interpolation[66]. The values in Kelvin were converted to Celsius.

As with previous data, 16 years of daily surface solar radiation downwards data, which is the sum of shortwave radiation reaching the surface of the Earth, were also downloaded from the ERA-Interim. These data were generated from the ECMWF forecast atmospheric model with a T255 horizontal resolution (~80 km) and vertical resolution of 60 model layers[66].

Similarly, 16 years of daily photoperiod data, representing the length of time that a plant receives sunlight in a day, were also generated for this analysis. These data were estimated using a standard equation based on latitude and day of year[67] in the 'geosphere' package in R[68]. A summary of all datasets used in this research is shown in Table 1. All datasets were aggregated to match the spatial resolution of the temperature data (i.e., 0.125°).

The preseason periods used in this analysis were defined as the period preceeding the LSP event from 0 to 90 days in a step of 10 days. The sum of each time-series of gridded climatic data (rain, solar, temperature, and photoperiod) in a preseason period were summed up (10 days period) before each start of season (SOS) date and end of season (EOS) date for each year, and before median SOS and EOS dates of the time period.

**LSP drivers**. To investigate the relationship between each LSP parameter (SOS and EOS) and a single preseason climatic driver, we developed two statistical models; a simple linear regression and a partial correlation analysis. The partial correlation analysis measures the correlation between two variables after purposefully controlling the effects of other variables. This method has been used commonly in several remote sensing studies determining the drivers of LSP[6,24,69]. Due to the dominance of different climatic factors across different geographical areas, and the species-specific responses of vegetation to climatic factors[70], different models were developed and applied for different regions and vegetation types. These regions were chosen for their unique climatic features (Fig. 9)[57]. Furthermore, caused by the reported effects of SOS on EOS[69], to isolate the influence of SOS, dates of SOS were included in the partial correlation analysis as a variable in the EOS models. We also applied linear regression to the preseason sum of climatic data estimated from the median SOS and EOS values to obtain trends over the time period. The significance of the correlations at 95% confidence level was assessed for all models and for each vegetation type in the different geographical regions.

**Reporting summary**. Further information on research design is available in the Nature Research Reporting Summary linked to this article.

## Data availability

All data used in this work are publicly available via the sources which have been referenced in the methods section. The LSP data can also be assessed via https://doi.org/10.5258/SOTON/D0407.

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

## Acknowledgements

We thank the Commonwealth Scholarship Commission in the UK for the funding provided to Tracy Adole, and the Ministerio de Ciencia, Innovación y Universidades of Spain (grant agreement RTI2018-096561-A-100) for supporting Victor Rodriguez-Galiano.

## Author contributions

T.A., J.D., V.R.G., and P.M.A. conceived and designed the research. T.A. performed the data analysis, and T.A., J.D., V.R.G., and P.M.A interpreted the data and wrote the paper.

## Competing interests
The authors declare no competing interests.
