## [Peer Review File · Communications Biology]

Reviewers' comments:

Reviewer #1 (Remarks to the Author):

Tracy Adole et al., aimed to investigate the relationship between climatic variables (e.g. temperature, precipitation, solar insolation, photoperiod) on vegetation phenology across Africa. Based on satellite-derived Start/End of season (SOS/EOS), the authors found that photoperiod is the dominate factor while other variables also contribute to the changes in vegetation phenology. Such result sounds interesting and has its own merits. However, several serious concerns on the method and result section (refer to my major issues), especially for the determination of "preseason" and the analysis of photoperiodic effect, are required to be carefully addressed before possible acceptance. A revision major is therefore recommended.

Major issues:

First, the authors summarized climatic variables during the period prior to SOS/EOS following the concept of "preseason" (Lines: 163-168). Then, why the authors applied 30 days preseason across the entire analysis (Figures 1-3 and 5-8)? Did all pixels share the same "preseason"? I have also noticed Tables. 2-5, the correlations between climatic variables and SOS/EOS depends heavily on the choice of "preseason". I am not sure if 30 days preseason should be regarded as the optimal season.

Second, the authors attempted to consolidate the relationship between photoperiod and vegetation phenology across the Africa. However, the authors did not provide the spatial patterns of changes in SOS/EOS, climatic variables. Correlation analysis alone might be not enough to explain changes in vegetation phenology. The authors suggested that a combination of climatic variables exerted controls on vegetation phenology, therefore, the spatial distribution of dominate climatic factors (temperature, precipitation, insolation and photoperiod) might be more interesting.

Third, the authors investigated the impact of climatic variables on SOS/EOS at different vegetation types and regions (e.g. extreme North, Northern Africa and Southern Africa), while there were no related figures providing us such information. In addition, the authors focused their study on cropland. Then, how did the authors deal with the "multiple growing seasons" in the extraction of SOS/EOS?

Fourth, the scatter plots between climatic variables and SOS/EOS (e.g. Figures 1 and 2) are a bit strange. I am wondering how the authors deal with the spatial mismatch between climatic variables and SOS/SOS, since they have different spatial resolutions.

Minor issues:

-Lines: 25-27. Please specify the study region to make the statement more precise.

-Line: 156. What does "a T255 horizontal resolution" mean?

-Lines:283-292. Table 4 is not well presented, please revise it.

-Figure 9. The regions covered with non-vegetation should be marked with another color (i.e. not grey). Also, the percentage of regions expressed significant changes in SOS/EOS?

-Figures S1 and S2. The authors should revise the title to clarify the period when maximum and minimum photoperiod was calculated and why these two variables were required there.

Reviewer #2 (Remarks to the Author):

This study used the satellite data and reanalysis meteorological data to analyze the relationship between vegetation phenological parameters and different climatic drivers to provide increased understanding of vegetation phenology of Africa. However, the manuscript seems simple and

insightful discussion is needed. This article needs a major revision before it is acceptable.

The accuracy of phenological metrics depends on a variety of factors including the quality and temporal resolution of satellite data, more introduction about the data limitations, such as imposed by cloud over, should be added.

In Fig 1-3, how to differentiate the different parts, as northern hemisphere, southern hemisphere, extreme northern part of Africa? And why did they differentiate into three parts? Not 2 or 4 parts. I would suggest to add a geographic map to show different study regions.

A brief introduction about the study area is needed in the data and methodology.

Table 3, also for other figures, why the partial correlation coefficients between SOS of the croplands and photo are all 0.96, and the partial correlation coefficients between SOS of the grasslands and photo are all 0.98? Is it a writing mistake? I suggest change the "*" to other icon, for the star always used for significant and is prone to misunderstand when read the Tables.

Table 4, there are some display errors.

Most of the discussion part is simply replying on the results and little comparison with other studies, I suggest to add more details about the eco-physiological interpretation, comparison with in situ measurements and comparison with global results.

Photoperiod controls vegetation phenology across Africa

Response to reviewers

August 2019

Response to Reviewer #1

Comment 1: *First, the authors summarized climatic variables during the period prior to SOS/EOS following the concept of “preseason” (Lines: 163-168). Then, why the authors applied 30 days preseason across the entire analysis (Figures 1-3 and 5-8)? Did all pixels share the same “preseason”? I have also noticed Tables. 2-5, the correlations between climatic variables and SOS/EOS depends heavily on the choice of “preseason”. I am not sure if 30 days preseason should be regarded as the optimal season.*

Response: Thank you for this useful remark. We agree that the 30 days preseason shouldn't be regarded as the optimal season, and we wish to state that this was never inferred in the study. Additionally, all preseason; 10 – 90 days in the steps of 10 were applied across the entire analysis as shown in the Supplementary Table 1-6, however only the 30 days plots were shown as it would be impossible to show all plots for all 9 steps in the manuscript. Nevertheless, in order to make this very clear figure captions have been reviewed to reflect the fact that the 30 days preseason period was just an example plot.

Scatterplots between SOS and climatic drivers cumulated over a 30 day preseason period across different vegetation types in the Northern hemisphere of Africa (All at $P < 0.05$). Plots for other preseason periods are not shown (see Table *). Shrublands were located in the horn of Africa

Comment 2: *Second, the authors attempted to consolidate the relationship between photoperiod and vegetation phenology across the Africa. However, the authors did not provide the spatial patterns of changes in SOS/EOS, climatic variables. Correlation analysis alone might be not enough to explain changes in vegetation phenology. The authors suggested that a combination of climatic variables exerted controls on vegetation phenology, therefore, the spatial distribution of dominate*

climatic factors (temperature, precipitation, insolation and photoperiod) might be more interesting.

Response: Thank you for this very good observation. We understand that the spatial pattern of changes in SOS, EOS and climatic variable is important as indicated by the reviewer. Hence, we added the figure to show the spatial pattern of changes in SOS/EOS also referred to as the inter-annual trends in Figure 8. However, our focus here was not to estimate the inter-annual trends in all the climatic variables. This is because it would require a substantial longer time period to determine either what pre-season period would be best to investigate or what monthly/yearly average inter-annual trends to investigate. This essentially would warrant a comprehensive new study, beyond the main scope of this current, already extensive work. Thus, we decided to only show the results of the spatial changes of 30 days pre-season temperature because these were more significantly correlated to SOS/EOS than any other climatic factor except photoperiod. It is important to note that photoperiod is consistent year-to-year. Nevertheless, we provided the yearly mean of all climatic factors in the supplementary data.

Supplementary Figure 3: Spatial distribution of 2016 yearly mean values of climatic factors

Comment 3: *Third, the authors investigated the impact of climatic variables on SOS/EOS at different vegetation types and regions (e.g. extreme North, Northern Africa and Southern Africa), while there were no related figures providing us such information. In addition, the authors focused their study on cropland. Then, how did the authors deal with the “multiple growing seasons” in the extraction of SOS/EOS?*

Response: Thank you once again for this useful remark. In addressing this point, we have incorporated a new map showing the different regions and we have added the following statement below to the methods section:

Figure 9 Reclassified land cover types and the different geographical study regions.

the Discrete Fourier Transform (DFT) technique was used to estimate the Start of Season (SOS) and End of Season (EOS) parameters for each year, for each image pixel covering the period 2001 to 2015. This technique is particularly useful in detecting the bimodal seasonality and double cropping agricultural system found in some parts of Africa. However, only the first season was considered in this analysis due to the very limited occurrence of the second season and limited spatial coverage of the corresponding climatic data.

Comment 4: *Fourth, the scatter plots between climatic variables and SOS/EOS (e.g. Figures 1 and 2) are a bit strange. I am wondering how the authors deal with the spatial mismatch between climatic variables and SOS/SOS, since they have different spatial resolutions.*

Response: Thank you for this useful observation. Not adding the approach use to deal with the spatial mismatch was indeed a major oversight and this has been addressed with the following:

...all datasets were then aggregated to match the spatial resolution of the temperature data (i.e. 0.125°).

Comment 5: -Lines: 25-27. Please specify the study region to make the statement more precise.

Response: Thank you for this suggestion. We have added the phrase “the different regions” to reflect the fact again that it was in all the regions of the African continent that photoperiod was seen to be the dominant factor.

Comment 6: -Line: 156. What does “a T255 horizontal resolution” mean?

Response: Many thanks for this question. T255 horizontal resolution refers to the triangular horizontal resolution in wave numbers that was used to develop the ECMWF data. This is equivalent to 80 km. More details can be found here <https://www.ecmwf.int/en/forecasts/documentation-and-support/changes-ecmwf-model/ifs-documentation>

For clarity we have added (approximately 80 km) to the sentence.

Comment 7: -Lines:283-292. Table 4 is not well presented, please revise it.

Response: Thank you for this observation; this has been rectified.

Comment 8: -Figure 9. The regions covered with non-vegetation should be marked with another color (i.e. not grey). Also, the percentage of regions expressed significant changes in SOS/EOS?

Response: Thank you once again for this query. We tried to use other colours but discovered that the grey background was the best contrast achieved in showing the significant pixels. See both figures below:

Comment 9: -Figures S1 and S2. The authors should revise the title to clarify the period when maximum and minimum photoperiod was calculated and why these two variables were required there.

Response: Thank you for this valuable suggestion. We have added the following to the figure captions "*for the period 2001-to-2015*". The maximum and minimum photoperiod were calculated to show the relationship between different sums of photoperiod and LSP parameters, since photoperiod was the most dominate climatic factor associated with LSP parameters.

Response to Reviewer #2

Comment 1: *The accuracy of phenological metrics depends on a variety of factors including the quality and temporal resolution of satellite data, more introduction about the data limitations, such as imposed by cloud over, should be added.*

Response: We thank you for your interest and constructive review of this study. We have added the following to the discussion section as you suggested.

Another possible limitation of this study relates to uncertainties in the remotely sensed data. These range from sensor degradation issues to saturation effects due to clouds and aerosols⁵⁷ that can affect the quality of the data and in turn the phenological variable. However, steps were taken to mitigate the effect of such limitations, including computing the EVI (see methods section) to reduce the effect of saturation.

Comment 2: *In Fig 1-3, how to differentiate the different parts, as northern hemisphere, southern hemisphere, extreme northern part of Africa? And Why did they differentiate into three parts? Not 2 or 4 parts. I would suggest to add a geographic map to show different study regions.*

Response: We appreciate this suggestion and have added a map showing the different parts to the manuscript. Ideally, at the continental scale analysis one would expect two sections: northern hemisphere and southern hemisphere to account for different climatic patterns. However, in this cases we added an extra group as extreme north to separate the vegetation types in the north and south of the Sahara desert.

Figure 9 Reclassified land cover types and the different geographical study regions.

Comment 3: *A brief introduction about the study area is needed in the data and methodology.*

Response: Thank you for this very insightful remark. We agree with this and have now revised the methods section to reflect it. See below:

Africa is the second largest continent in the world covering an area of over 30 million km², and its latitude ranges from 37°N to 35°S with the Equator in the middle⁵⁸. With a diverse range of vegetation types as shown in Figure 9, Africa is home to the world largest area of savannah, the second-largest rainforest in the world, and the second and third largest wetlands in the world⁵⁹. Africa has a distinctive pattern of climate which varies by geographical sub-region and which plays a significant role in the vegetation dynamics of Africa⁵⁸.

Comment 4: *Table 3, also for other figures, Why the partial correlation coefficients between SOS of the croplands and photo are all 0.96, and the partial correlation*

coefficients between SOS of the grasslands and photo are all 0.98? is it a writing mistake? I suggest change the "" to other icon, for the star always used for significant and is prone to misunderstand when read the Tables.*

Response: Thank you for this very valuable comment. The partial correlation coefficients are actually correct and not a writing mistake. Also, we agree that the use of an asterisk to mean insignificant can be misunderstood. Rather than change the symbol we have reversed the use of it. We have used it on all values that are significant. This is because after an extensive search of several documents/studies, there is no better symbol to use than the asterisk when trying to make a reference or distinguish the meaning of numbers in a report, and this is the standard practice.

Comment 5: *Table 4, there are some display errors.*

Response: Thank you for this observation; this has been rectified.

Comment 6: *Most of the discussion part is simply replying on the results and little comparison with other studies, I suggest to add more details about the eco-physiological interpretation, comparison with in situ measurements and comparison with global results.*

Response: Thank you for this very valuable suggestion. However, we wish to state here that a thorough literature review was done to compare results with other studies, and these studies have already been referred to extensively in the discussion. The reason more comparison was not done is because very few studies have looked at the effect of photoperiod on vegetation phenology, especially in the tropics. We also wish to state here that there are very few *in situ* measurements that have been done in Africa, most of which were also referred to in this study. In fact, some major issues highlighted in this study are **insufficient ground observation** and **the need to incorporate photoperiod into phenological models** as these are currently lacking in phenological studies. We also wish to highlight areas in the discussion that have addressed some of the points raised.

Eco-physiological interpretation: *It has been established that C4 plant types are better adapted to warm climates because of their enzyme sensitivity to chilling*

temperatures⁵³. They are also known to have greater photosynthetic capacity at higher sunlight and temperature levels⁵⁴... This distinct change in the response of plants to small changes of 2 hours or less in photoperiods has been reported previously^{46,64}. Likewise, it has also been suggested that plants respond to specific critical daylength (varies from plant to plant) during which hormonal regulation of growth initiation or cessation hormones occurs^{66,67}. This ability of plants to detect light and measure time very accurately has been attributed to an "endogenous time-keeping mechanism called the circadian clock"⁶⁶, and the perception of light signals by photoreceptors³, and these clock and photoreceptors genes can be found in all living plant cells⁶⁷.

In-situ measurements: This sensitivity has been reported for a wide range of vegetation types in the Sudanian region⁴⁷ and is also known to be genetically-based in some cereal crops, including major varieties grown in West Africa^{48,49}... These conditions are mostly favourable to tropical plants, known to grow well in warmer temperatures and with shorter photoperiods⁴³, particularly millet and sorghum, two of the most important cereal crops grown in Sudano-Sahelian region⁵⁰... Studies have shown that the grasses (mainly of C4 type) found in many African ecosystems are associated with high solar radiation and temperatures⁵²... These responses to photoperiod have been shown to significantly influence the population structure of major crops like millet and sorghum in Western and Central Africa than any other environmental factor, a sole factor for adaptation to environmental constraints^{50,68}... This distinct response of crops to the direction of photoperiod also reflects the results from⁵⁵ who found out that leafing rate was determined by the duration and direction of photoperiod at seed germination.

Global results: This dominant control by photoperiod tends to corroborate earlier work which attributes photoperiod relative to some threshold to be the major determinant that allows other climatic-driven development to occur⁴².... The results are also in agreement with earlier studies indicating that a combination of climatic factors, either occurring simultaneously or preceding one another, controls LSP patterns, with their effects sometimes biome-dependent^{1,2,43}... This result for Africa is nevertheless consistent with other research which concluded that photoperiod is the major factor controlling phenological events in tropical ecosystems^{22,46}.... These findings are consistent with⁵⁷ and corroborate the idea that photoperiod is the most reliable predictor of onset dates for southern African savanna trees^{58,59}... These

results confirm previous suggestions that a combination of photoperiod and temperature thresholds are environmental cues for vegetation growth in southern Africa ²¹... Irrespective of the observed dominance of photoperiod, the partial correlation results also showed that LSP is influenced not just by a single factor but by a combination of these factors, which is in agreement with previous studies showing that most phenological phases are controlled by both photoperiod and temperature ^{1,2}.

REVIEWERS' COMMENTS:

Reviewer #1 (Remarks to the Author):

I have reviewed the changes made by the authors. I am satisfied. The paper may be accepted for publication.

Reviewer #2 (Remarks to the Author):

This study used the satellite data and reanalysis meteorological data to analyze the relationship between vegetation phenological parameters and different climatic drivers. I think this version has been improved much, but still need a minor revision before it can be accepted.

- 1) Spelling mistakes still existed. It has to be polished.
- 2)The discussion section is lengthy and complicated. It needs further work to make it concise.